# Validation of a Novel Method to Evaluate Community-Based Interventions That Improve Access to Fruits and Vegetables

**DOI:** 10.3390/ijerph22020312

**Published:** 2025-02-19

**Authors:** Louisa Ewald, Kate E. LeGrand, Claire-Lorentz Ugo-Ike, Sally Honeycutt, Jennifer L. Hall, Emmanuela Gakidou, Ali H. Mokdad, Gregory A. Roth

**Affiliations:** 1Institute for Health Metrics and Evaluation, Seattle, WA 98105, USA; louisa13@uw.edu (L.E.); kel15@uw.edu (K.E.L.); gakidou@uw.edu (E.G.); mokdaa@uw.edu (A.H.M.); 2American Heart Association, Dallas, TX 75231, USA; claire.ugo-ike@heart.org (C.-L.U.-I.); sally.honeycutt@heart.org (S.H.); jennifer.hall@heart.org (J.L.H.)

**Keywords:** community-based programs, social determinants of health, fruit consumption, vegetable consumption, dietary behaviors, survey methods

## Abstract

Background: Current evaluation tools are inadequate for assessing the impact of small-scale interventions, such as farmer’s markets or community meal programs, on fruit and vegetable consumption. This study analyzes the pilot data of a novel tool designed to evaluate community-based programs’ impact on fruit and vegetable consumption. Our research addresses the gap in effective evaluation methods for dietary behaviors within underserved populations. Methods: The survey tool was developed through a participatory research approach involving interest holders and community members. We conducted a pilot survey across four community-based programs, validated the findings, and compared them against the data from the Behavioral Risk Factor Surveillance System. Results: This pilot survey demonstrated a high completion rate of 98.2%. Notably, 62.5% of respondents reported an increased consumption of fruits and vegetables since participating in the programs and cited cost, time, and accessibility as primary barriers to healthy eating. There is a strong, though not significant, correlation of 0.876 (*p* = 0.12) between the pilot data of prevalence of daily fruit and vegetable consumption and the national average. Conclusion: Our findings suggest the survey tool effectively captures dietary behaviors and the influence of community-based programs. Further research is required to enhance its applicability in diverse settings and extend robust impact evaluation methods for these programs.

## 1. Introduction

Eating fruits and vegetables is essential for preventing cardiovascular diseases (CVDs), with the extensive research demonstrating their critical role in maintaining cardiovascular health [1,2]. The Global Burden of Diseases, Injuries, and Risk Factors Study (GBD) and other analyses on the balance of plant-based foods in diets have consistently highlighted fruit and vegetable consumption as important health metrics directly linked to reduced CVD risk [3]. Despite this, the challenge of effectively promoting and measuring healthy eating behaviors, particularly in underserved communities, remains an important public health challenge [4,5,6].

Community-based food access programs have emerged as vital interventions aimed at improving dietary habits and enhancing community health outcomes [7,8,9]. These programs, such as farm-to-consumer produce boxes and mobile food delivery services, seek to address the barriers to healthy food access, including physical availability, affordability, and quality of nutritious foods. The tools used to assess fruit and vegetable intake generally include a food frequency questionnaire (FFQ) consisting of asking the consumption frequency of up to 120 food items, a 24 h recall, or diet records [10,11]. These tools are not widely applied in diverse community settings as they are difficult to implement and have barriers to participation. Outside of rigorous studies with few participants, high costs, and low accessibility, there is a need for more innovative approaches to measure the effectiveness of programs [12]. Operating within these real-world contexts, community-based programs provide valuable insights into these challenges. In particular, food swamps, or areas where unhealthy food options dominate, highlight the barriers and challenges faced by underserved communities. Previous research and firsthand accounts have revealed common lived experiences, including the need for improved accessibility, affordability, quality, and healthier options, especially for families with children or adolescents [13,14,15]. Both the design and evaluation of such programs must focus on health equity and be inclusive, empathetic, and responsive to the needs of the communities they aim to serve [16]. Participatory research methodologies, which engage community members and stakeholders interest holders throughout the research process, offer a promising avenue for developing and implementing evaluation tools for community-based food access programs. These methodologies emphasize the importance of a collaboration between researchers and communities, recognizing the unique insights and experiences that community members bring to the research [17]. By involving community stakeholders interest holders in the development, testing, and refinement of evaluation tools, researchers can ensure that these instruments are not only relevant and culturally sensitive but also capable of capturing the nuanced impacts of community-based food access programs on dietary behaviors.

Designing interventions for community-based programs must emphasize the importance of not only short-term effectiveness, but also long-term resilience and the sustainability of the communities and their environments. To address this need, our study aims to answer the following research questions: How can a brief survey tool effectively measure the impact of community-based food access programs on fruit and vegetable consumption in underserved communities? And what barriers do participants in these programs face in increasing their consumption of fruits and vegetables, and how can these insights contribute to enhancing the effectiveness of such programs? By addressing these questions, our study contributes to filling the existing research gap by providing a tailored approach to an evaluation that reflects the complex interconnections between dietary behaviors, public health, and sustainable food systems.

This current study aimed to develop and pilot a brief survey tool specifically designed to assess fruit and vegetable intake in various community settings [18]. This tool was developed through a participatory research process, engaging key interest holders, including funders, program staff, public health experts, and community members themselves. By achieving face validity from these interest holders, content validity from experts, and construct validity through comparative analysis, our study sought to create a survey instrument that meets rigorous scientific standards and resonates with and is accessible to the diverse populations served by community-based food access programs.

## 2. Materials and Methods

### 2.1. Survey Development and Validation

Our cross-sectional survey aimed to capture detailed information on fruit and vegetable consumption behaviors, barriers to healthy eating, and the impact of participation in community-based food access programs. This survey included sections on demographic indicators, program participation, and frequency of fruit and vegetable consumption (Appendix A). Additionally, it explored whether participation in these programs influenced the participants’ consumption habits and identified the main factors driving any reported changes.

The survey’s design drew from a thorough review of existing dietary assessment tools and relevant literature, aiming to address the unique challenges of evaluating fruit and vegetable intake in diverse community settings [19,20,21]. These sources were identified through a search of the academic literature and consultations with public health experts. This process highlighted the importance of developing a tool that could navigate the socioeconomic and cultural nuances of the target population.

To ensure the survey was both inclusive and accessible, questions were translated into Spanish by a native speaker and underwent a review for cultural appropriateness by program staff. The survey prioritized clarity and brevity to facilitate understanding and encourage participation, thereby generating reliable data for meaningful insights.

Interest holder feedback was integral to refining the survey. We interviewed funders, program staff, and public health experts to validate the clarity, relevance, and comprehensiveness of the survey questions. Program staff and experts were able to suggest and review the entire questionnaire and collection methods during the development stage and community members were asked about their thoughts at the time of collection. The survey also included an open text field for community members to provide feedback on the tool. This additional input offered valuable insights into the survey’s feasibility and validity, affirming the survey’s relevance and accessibility to those it seeks to engage.

### 2.2. Pilot Test Procedure

The pilot test of the survey was conducted in April 2024 across four distinct settings selected by the American Heart Association’s Social Impact Funds. These settings included a free meal truck in Clearwater, Florida (Program A); a traditional farmer’s market stand in Atlanta, Georgia (Program B); a mobile farmer’s market in Antioch, California (Program C); and a farm bag program in Richmond, California (Program D).

Participants were recruited on-site during their attendance at one of the programs. They could choose to complete the survey on a tablet, smartphone, or through a read-aloud option provided by the researchers to accommodate different preferences and abilities. Upon completion, participants were offered an incentive of USD 10 worth of produce to redeem with the program.

All study methods, including recruitment strategies, informed consent procedures, and participant incentives, were reviewed and approved by the University of Washington Institutional Review Board (STUDY00020152). Participants were informed about the study’s purpose before consenting to participate, and they had the option to stop the survey at any time.

### 2.3. Measures

The measures in this study were designed to capture fruit and vegetable consumption behaviors, as well as to identify barriers to healthy eating and the impact of participation in community-based food access programs.

The survey included demographic questions to gather information on the participants’ age, gender, race, and ethnicity, providing context for analyzing consumption patterns across different demographic groups. A question on household food security helped to understand the participant’s ability to afford food [22]. This question asked participants about their food security, and answer choices included having enough food at home, having enough food, but not the kinds they want to eat, sometimes not having enough, or often not having enough.

Questions related to fruit and vegetable consumption asked participants to report the frequency of their intake, ranging from five or more times per day down to a few times a month, or never [21]. The survey also included descriptions of what is considered a fruit and vegetable, which included fresh or frozen but not canned or processed items. Both the unit of measure and description align with the Behavioral Risk Factor Surveillance System (BRFSS) fruit and vegetable module. Distinguishing between fruits and vegetables is crucial for understanding dietary habits and aligns with the dietary guidelines that recommend specific intake levels for each.

The survey also asked about participants’ consumption patterns before and after becoming participants in the programs. Participants were able to specify whether their consumption since participating in the program increased a lot, increased a little, stayed the same, or decreased. These questions were designed to evaluate whether participation in these programs influenced the participants’ fruit and vegetable consumption, and reasons for any change in consumption.

Barriers to healthy eating were explored by allowing participants to identify the challenges they faced in consuming more fruits and vegetables. These included options related to cost, accessibility, lack of knowledge on preparation, lack of time for preparation, taste preferences, and availability of preferred types and varieties.

In addition to the quantitative measures, open-ended questions were included to gather qualitative data on participants’ experiences with the program, attitudes towards fruits and vegetables, and suggestions for program improvement. This qualitative component enriched the quantitative findings, providing a more nuanced understanding of the factors influencing fruit and vegetable consumption and the potential impact of community-based food access programs.

### 2.4. Data Analysis

Our analysis used quantitative data to assess the survey’s validity and reliability. Descriptive statistics provided an overview of the collected data, while the survey’s face and content validity were evaluated through expert consultations and feedback from the pilot test. Criterion validity was established through a comparison with the Behavioral Risk Factor Surveillance System (BRFSS) fruit and vegetable module. Specifically, we used Pearson’s correlation coefficient to compare the prevalence of consuming at least one fruit or vegetable daily against this recognized public health surveillance standard.

Qualitative data, primarily from the open text responses, were categorized manually due to the small data size and focus on quantitative data. Comments were categorized by positive and negative sentiments, comments regarding the program delivery, and comments regarding the survey or collection method itself.

### 2.5. Feasibility 

Feasibility was assessed based on the survey completion rate, administration method, and participant feedback on the survey process. This evaluation aimed to determine the practicality of deploying the survey tool in diverse community settings and its ability to engage the target population effectively.

## 3. Results

### 3.1. Participant Demographics

The study gathered data from 111 survey respondents across the four programs included in the pilot. The age distribution of participants, presented in Table 1, showed a predominance of individuals aged 30–49 years, accounting for 44% of the total, followed by those aged 50–69 years (31.2%), 18–29 years (14.7%), and 70+ years (10.1%). In terms of gender, the majority of respondents was female (83.8%), with males representing 16.2% of the sample. Racial and ethnic diversity were also captured, with White, non-Hispanic participants forming the largest group at 40.5%, followed by Black, non-Hispanic (24.3%), Latino/a (15.3%), Asian (8.1%), multi-racial (7.2%), Pacific Islander (0.9%), and a small percentage (3.6%) preferring not to respond. Regarding food security, a majority (68.9%) reported having enough of the kinds of food they wanted to eat, while 19.8% had enough but not always the kinds they preferred, and 11.3% sometimes or often did not have enough food to eat.

### 3.2. Survey Completion Rate and Administration Method

The survey demonstrated a high completion rate of over 98.2%, indicating that nearly all participants were able and willing to complete the survey once they started.

Over half (55.9%, n = 62) of the participants took the survey on a tablet and spent an average of 8.3 min on it, while 44.1% (n = 49) of participants took the survey on a smartphone and spent on average 5.1 min to complete the survey.

### 3.3. Feasibility and Acceptability

In terms of participant experience taking the survey, many expressed gratitude and positivity towards the programs and were grateful for the survey incentive. Several people commented that the survey was “easy to do”, and one person commented that they were “glad to be taking this survey” and that “it should be done more often to gather data on population habits”. Administering the survey in person was important as some participants were uncomfortable with navigating the survey themselves on the tablet and asked to have the survey read aloud to them.

### 3.4. Fruit and Vegetable Consumption

Participants reported eating fruits an average of 1.6 (1.3 SD) times a day and eating vegetables an average of 1.9 (1.4 SD) times a day. Figure 1 shows the distribution of answers, with over half (59.6%, n = 65) of participants saying that they eat fruit at least once per day and 5.5% (n = 6) eating fruit five or more times per day. Participants reported a higher consumption of vegetables, with 71.6% (n = 78) eating vegetables at least daily, with 7.3% (n = 8) eating vegetables five or more times per day.

Along with the current frequency of fruit and vegetable consumption, the survey asked participants whether the consumption of fruits and vegetables increased, decreased, or stayed the same since becoming involved in the program. Over a third of respondents (37.5%, n = 33) said that their consumption increased a lot, with an additional 25% (n = 2) saying that their consumption increased a little. Another 37.5% (n = 33) said that their consumption stayed the same, and zero participants said that their consumption decreased since becoming involved in the program.

### 3.5. Barriers

In exploring the barriers faced by participants in consuming more fruits and vegetables, the survey revealed a variety of challenges for people hoping to consume more fruits and vegetables in their diet, as seen in Figure 2. Cost emerged as the most prominent barrier, with half of the respondents (50%, n = 47) indicating that high prices hindered their fruit and vegetable intake. This was followed by a lack of time for preparation, cited by 23.4% (n = 22) of participants, underscoring the impact of busy lifestyles on healthy eating habits. Accessibility issues, including distance to stores and transportation difficulties, were also notable, affecting 19.1% (n = 18) of the survey population.

Other barriers identified included a limited variety of fruits and vegetables available, which 18.1% (n = 17) of respondents found problematic. This was closely aligned with the 17% (n = 16) who selected the “Other” category, suggesting unique or unspecified challenges not directly listed in the survey options. Additionally, 9.6% (n = 9) of participants pointed to a lack of knowledge on how to prepare fruits and vegetables, indicating a potential area for educational interventions.

Few participants reported taste preferences and a predilection for other foods as major obstacles, with only 2.1% (n = 2) and 5.3% (n = 5), respectively, indicating these as barriers. Not finding the specific kinds of fruits and vegetables they wanted was a barrier for 7.4% (n = 7) of respondents, highlighting the importance of variety in promoting fruit and vegetable consumption.

### 3.6. Criterion Validity

In assessing the criterion validity of the pilot survey data on fruit and vegetable consumption, we observed a strong positive linear relationship with the data from the Behavioral Risk Factor Surveillance System (BRFSS), indicated by a correlation coefficient of 0.876, though not statistically significant (*p* = 0.124) due to the insufficient sample size. This strong correlation suggests that our survey can effectively capture dietary behaviors in a manner consistent with the established BRFSS standard.

Figure 3 represents the prevalence of daily or more fruit and vegetable consumption among participants of the community-based food access programs, compared to the BRFSS national average. For fruits, the data points for each of the four community programs show varying levels of fruit consumption prevalence, with Program D reporting the highest average daily consumption at 80%, and Program A reporting the lowest average at 45.5%. The national average for consuming fruit at least once daily, as shown by the BRFSS data, is 60.1%.

For vegetables, Program D displays a similar trend at an 80% consumption prevalence, with two programs showing the lowest prevalence of 63.6%. The BRFSS average for vegetable consumption is 79.4%, positioning it higher than most of the program-specific rates, except for Program D, which matches the BRFSS average.

## 4. Discussion

Our study aimed to develop and pilot test a brief survey tool assessing fruit and vegetable intake among participants of community-based food access programs. The findings from our pilot study underscore the tool’s effectiveness in capturing the relevant data on fruit and vegetable consumption, with a high completion rate suggesting its accessibility and relevance to the target population. Comparatively, similar studies have demonstrated the necessity of accessible evaluation tools in understanding the impact of nutritional interventions, yet few have provided a clear methodology for engaging with community-based programs [19,23,24,25,26,27,28]. While these studies contribute valuable insights into public health nutrition strategies, they often overlook the unique dynamics and challenges faced by smaller, community-based food access programs. These hyperlocal, small-scale programs serve as touchpoints for individuals who might not typically engage in traditional research efforts.

Notably, 62.5% of participants reported an increase in their consumption of fruits and vegetables since becoming customers of their respective programs, highlighting the potential impact of these community-based interventions. The fact that zero participants reported a decrease in consumption further strengthens the argument for the positive influence of these programs. This observed increase is in line with the findings from other interventions, which report significant improvements in fruit and vegetable consumption among low-income populations through similar community-based programs [8,11,27,29].

Our work differs in its targeted approach, focusing specifically on these smaller settings and their unique participant demographics. This focus is essential for developing effective, scalable solutions tailored to the needs of communities traditionally underserved or left out of the dietary behavior research. Through adopting a participatory research method, our evaluation helps ensure cultural sensitivity and relevance, addressing a gap in the existing surveillance tools.

Our results also provide more comprehensive insights into the barriers to healthy eating, with cost and affordability emerging as the most important factor. This is consistent with the broader literature indicating that economic factors play a crucial role in dietary choices [25,29]. The correlation observed between the pilot data and the national averages from the BRFSS indicates the potential that our tool can effectively capture dietary behaviors in a manner consistent with the established public health surveillance standards. The program with the highest fruit and vegetable consumption rate, a discounted farm box program, suggests that affordability along with accessibility are important for overcoming barriers.

While national surveys like the BRFSS provide valuable population-level insights into dietary behaviors, they are not designed to evaluate the specific impact of interventions, particularly those tailored to small, community-based settings. Unlike large-scale national surveys, which collect broad epidemiological data, our survey tool is specifically developed to assess the direct effects of community-based food access programs on individuals’ fruit and vegetable consumption. The data from the BRFSS cannot measure the outcomes of localized interventions, nor are they feasible to implement within these specific contexts. Our tool, therefore, fills this gap by offering a means to assess the direct effects of community-based food access programs on dietary habits. This specificity allows for a more focused approach to addressing dietary challenges and also underscores the importance of creating evaluation tools that are as nuanced and targeted as the interventions they seek to assess.

Our approach seeks to engage individuals who typically fall outside the purview of conventional systems research, which often demands comprehensive and varied data inputs, including individual, community, environmental, and policy-related factors [24,30]. Recognizing the challenges associated with engaging these populations, our goal is to streamline the data collection process, making it swift and accessible enough to include a broader demographic. We aim to bridge the gap between the complex theoretical frameworks of systems research and the practical, lived experiences of individuals. This approach ensures that our research is grounded in the realities of those it intends to serve.

### Limitations

The pilot testing was conducted in a limited number of settings, which may affect the generalizability of our findings and may not capture the full range of experiences and perspectives. Additionally, the survey was administered at a single point in time. Extending the data collection period could potentially increase the sample size and diversity of responses, thereby improving the generalizability of the results. Additionally, the self-reported nature of the survey data may introduce selection bias. However, the survey tool was designed to be scaled to larger populations, thereby reducing the potential for bias and improving the ability to detect statistically significant results. Future research should aim to validate the tool across broader populations and explore the use of objective measures to complement self-reported data.

## 5. Conclusions

Our study presents a novel survey tool designed to assess fruit and vegetable consumption within the context of community-based food access programs. The initial validation suggests its potential as a reliable instrument for collecting data on dietary behaviors, offering valuable insights for interventions aimed at improving public health nutrition. The survey tool has potential applications beyond the initial pilot settings. It can serve as a valuable resource for funders and program administrators seeking to evaluate the impact of food access interventions. For policymakers, understanding the specific barriers to fruit and vegetable consumption can inform the development of targeted subsidies or support programs. Moreover, the tool’s adaptability makes it suitable for developing modules to assess other social determinants of health, further expanding its utility in public health research. Further research and refinement are needed to enhance its applicability and effectiveness in diverse settings.

## Figures and Tables

**Figure 1 ijerph-22-00312-f001:**
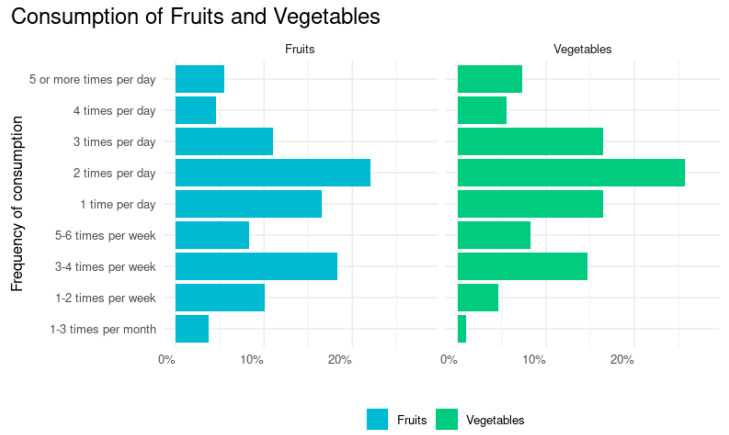
Frequency of fruit and vegetable consumption. Figure 1 displays the proportion of participants eating fruits and vegetables at various frequencies. Frequencies are in the range of 1–3 times a month to 5 or more times per day. No participants said that they never eat either fruits or vegetables. Data are from a pilot study of 111 participants across four community-based food access programs in April 2024.

**Figure 2 ijerph-22-00312-f002:**
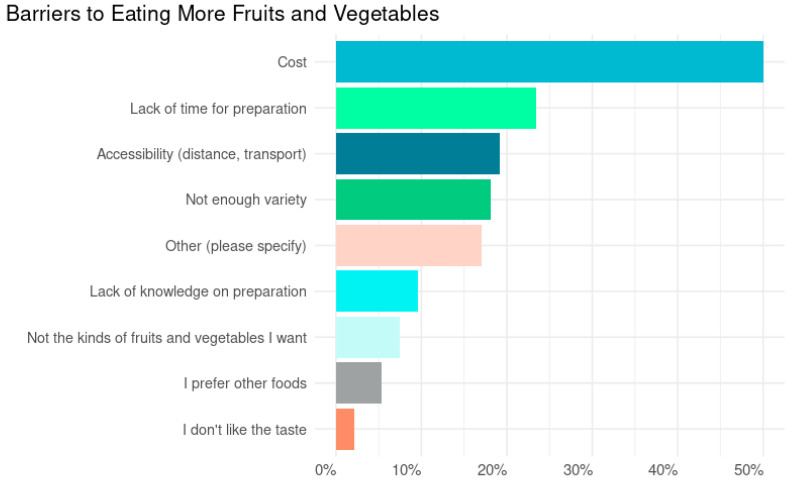
Barriers to eating more fruits and vegetables. Figure 2 displays the proportion of participants identifying barriers to eating more fruits and vegetables. Participants were able to select multiple barriers. Data are from a pilot study of 111 participants across four community-based food access programs in April 2024.

**Figure 3 ijerph-22-00312-f003:**
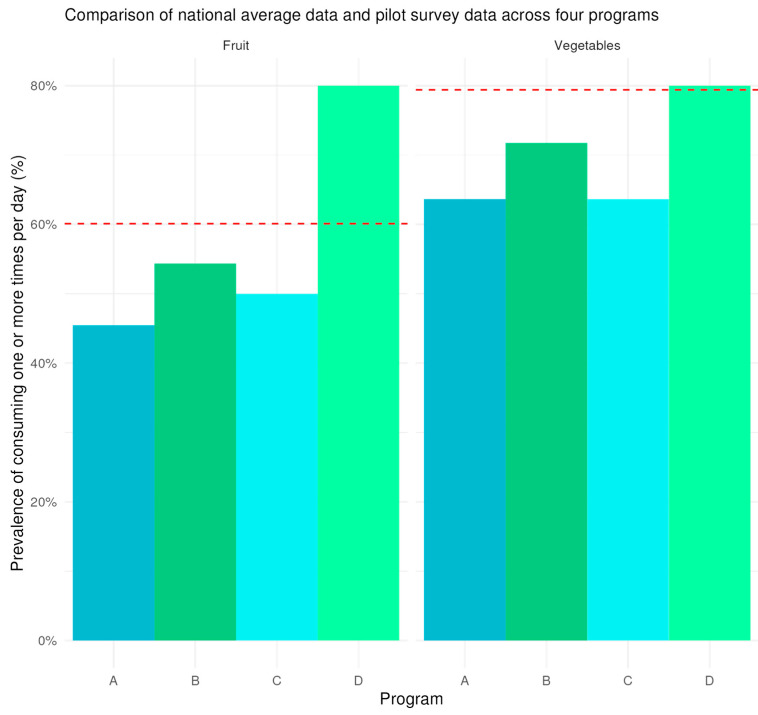
Fruit and vegetable consumption prevalence from the pilot survey and national-level data. Figure 3 displays the prevalence of participants in each program eating at least one fruit or vegetable daily compared to the national mean. National mean data come from the Behavioral Risk Factor Surveillance Survey (BRFSS) for 2023. Program data are from a pilot study of 111 participants across four community-based food access programs in April 2024.

**Table 1 ijerph-22-00312-t001:** Participant demographics.

Demographic Variable	n (%)
Age	
18–29 years old	16 (14.7%)
30–49 years old	48 (44%)
50–69 years old	34 (31.2%)
70+ years old	11 (10.1%)
Gender	
Female	93 (83.8%)
Male	18 (16.2%)
Race and ethnicity	
White, non-Hispanic	45 (40.5%)
Black, non-Hispanic	27 (24.3%)
Latino/a	17 (15.3%)
Asian	9 (8.1%)
Multi-racial	8 (7.2%)
Pacific Islander	1 (0.9%)
Prefer not to respond	4 (3.6%)
Food security	
Enough of the kinds of food I/we wanted to eat	73 (68.9%)
Enough, but not always the kinds of food I/we wanted to eat	21 (19.8%)
Sometimes or often not enough food to eat	12 (11.3%)

## Data Availability

Data from this study will not be made publicly available due to the IRB approval stipulations.

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
