# Peer review of "Validation of a Novel Method to Evaluate Community-Based Interventions That Improve Access to Fruits and Vegetables"

_ijerph, 2025, doi:10.3390/ijerph22020312_

Round 1
Reviewer 1 Report
Comments and Suggestions for Authors
The paper is interesting for the public health nutrition field and well-written. However, I have some remarks.
Introduction
I suggest adding literature concerning the file-a-day campaign promoted by the World Health Organization.
Methods/Results
The results are interesting, especially regarding the four programs, but the programs are not described, so the impact strength is unclear.
Discussion
The discussion should be widened according to the role of the programs.
Conclusion
The novelty and the usefulness of the tool are not stressed, please elaborate on this.
Author Response
Introduction
I suggest adding literature concerning the file-a-day campaign promoted by the World Health Organization.
Answer: Thank you, we have added this campaign as a reference in the introduction.
Methods/Results
The results are interesting, especially regarding the four programs, but the programs are not described, so the impact strength is unclear.
Answer: We have added more details in the methods section as to which program is which (i.e. Free meal truck in Florida is Program A).
“These settings included a free meal truck in Clearwater, Florida (Program A); a traditional farmer’s market stand in Atlanta, Georgia (Program B); a mobile farmer’s market in Antioch, California (Program C); and a farm bag program in Richmond, California (Program D).”
Discussion
The discussion should be widened according to the role of the programs.
Answer: We have expanded the discussion to incorporate more of the study’s findings as well as takeaways from specific programs.
“The program with the highest fruit and vegetable consumption rate, a discounted farm box program, suggests that affordability along with accessibility are important for overcoming barriers.”
Conclusion
The novelty and the usefulness of the tool are not stressed, please elaborate on this.
Answer: Thank you for this comment as it is very important to the entirety of our paper. We have edited the discussion, conclusion, and abstract to reflect the novelty and usefulness of the tool more.
Reviewer 2 Report
Comments and Suggestions for Authors
The paper offers an interesting perspective on validating a novel method designed to evaluate community-based interventions that improve access to fruits and vegetables. However, several aspects of the paper could be further refined.
The purpose of the paper is outlined but lacks precise formulation, particularly in the abstract. Clarifying the study's objectives would significantly enhance its focus and scientific contribution.
In the “Introduction” section, it would be helpful to clearly state the specific research questions being addressed and to articulate how the study contributes to filling the existing research gap. Moreover, the paper could delve deeper into the theoretical foundations of the study, particularly the models of the circular economy and sustainable development. Explaining how the proposed research tool aligns with these theories would increase the scientific value and contextual relevance of the paper.
Section 2 (Materials and Methods) lacks sufficient detail regarding the criteria for literature selection and the tools employed for analysis. Providing explicit information on the selection criteria (e.g., time range, data sources) would improve the credibility and transparency of the study's results. Furthermore, while the description of the process of creating the research tool is generally sound, it should include more technical details regarding validation and qualitative analysis.
The tables and graphs (e.g., Table 1 and Figure 1) are clear and well-presented, but the commentary on the data is somewhat limited. Expanding the interpretation of these results would help establish stronger links between the presented data, the study’s findings, and its conclusions.
The Discussion section would benefit from a broader comparative analysis with other studies to better contextualize the results within the existing body of literature. Including such comparisons would highlight the study's contribution and provide a more comprehensive understanding of its significance.
The conclusions, while relevant, are too general and do not fully capture the importance of the study's findings. Offering more specific practical implications, such as recommendations for public policy or guidelines for food program organizers, would enhance the paper’s impact and applicability.
Finally, I recommend standardizing the font in the Appendix and ensuring visual consistency across all figures to enhance the overall presentation.
Author Response
The paper offers an interesting perspective on validating a novel method designed to evaluate community-based interventions that improve access to fruits and vegetables. However, several aspects of the paper could be further refined.
The purpose of the paper is outlined but lacks precise formulation, particularly in the abstract. Clarifying the study's objectives would significantly enhance its focus and scientific contribution.
Answer: We have edited the abstract to make the study objective clearer.
“Background: Current evaluation tools are inadequate for assessing the impact of small-scale interventions, such as farmer’s markets or community meal programs, on fruit and vegetable consumption. This study analyzes pilot data of a novel tool de-signed to evaluate community-based programs’ impact on fruit and vegetable consumption. Our research addresses the gap in effective evaluation methods for dietary behaviors within underserved populations. Methods: The survey tool was developed through a participatory re-search approach involving interest holders and community members. We conducted a pilot survey across four community-based programs, validated the findings, and compared them against data from the Behavioral Risk Factor Surveillance System. Results: The pilot survey demonstrated a high completion rate of 98.2%. Notably, 62.5% of respondents reported increased consumption of fruits and vegetables since participating in the programs and cited cost, time, and accessibility as primary barriers to healthy eating. There is a strong, though not significant, correlation of 0.876 (p = 0.12) between the pilot data of prevalence of daily fruit and vegetable consumption and the national average. Conclusion: Our findings suggest the survey tool effectively captured dietary behaviors and the in-fluence of community-based programs. Further research is required to enhance its applicability in diverse settings and extend robust impact evaluation methods for these programs.”
In the “Introduction” section, it would be helpful to clearly state the specific research questions being addressed and to articulate how the study contributes to filling the existing research gap. Moreover, the paper could delve deeper into the theoretical foundations of the study, particularly the models of the circular economy and sustainable development. Explaining how the proposed research tool aligns with these theories would increase the scientific value and contextual relevance of the paper.
Answer: We have added a paragraph to the introduction that explicitly states our research questions and delves deeper into the theoretical context of the research.
“Designing interventions for community-based programs must emphasize the importance of not only short-term effectiveness, but also long-term resilience and sustainability of the communities and their environments. To address this need, our study aims to answer the following research questions: How can a brief survey tool effectively measure the impact of community-based food access programs on fruit and vegetable consumption in underserved communities? And what barriers do participants in these programs face in increasing their consumption of fruits and vegetables, and how can these insights contribute to enhancing the effectiveness of such programs? By addressing these questions, our study contributes to filling an existing research gap by providing a tailored approach to evaluation that reflects the complex interconnections between dietary behaviors, public health, and sustainable food systems.”
Section 2 (Materials and Methods) lacks sufficient detail regarding the criteria for literature selection and the tools employed for analysis. Providing explicit information on the selection criteria (e.g., time range, data sources) would improve the credibility and transparency of the study's results. Furthermore, while the description of the process of creating the research tool is generally sound, it should include more technical details regarding validation and qualitative analysis.
Answer:
We have added a sentence describing sources and methods for reviewing literature and have also added a paragraph to the methods section addressing qualitative analysis.
“These sources were identified through a search of academic literature and consultations with public health experts.”
“Qualitative data, primarily from the open text responses, was categorized manually due to the small data size and focus on quantitative data. Comments were categorized by positive and negative sentiment, comments regarding the program delivery, and comments regarding the survey or collection method itself.”
The tables and graphs (e.g., Table 1 and Figure 1) are clear and well-presented, but the commentary on the data is somewhat limited. Expanding the interpretation of these results would help establish stronger links between the presented data, the study’s findings, and its conclusions.
The Discussion section would benefit from a broader comparative analysis with other studies to better contextualize the results within the existing body of literature. Including such comparisons would highlight the study's contribution and provide a more comprehensive understanding of its significance.
The conclusions, while relevant, are too general and do not fully capture the importance of the study's findings. Offering more specific practical implications, such as recommendations for public policy or guidelines for food program organizers, would enhance the paper’s impact and applicability.
Answer: Thank you for these recommendations to expand the interpretation and significance of the findings in the discussion. We have extensively edited the discussion to more explicitly compare the findings with other studies and provide more interpretation of the results. We have also stated more practical implications in the conclusion.
Finally, I recommend standardizing the font in the Appendix and ensuring visual consistency across all figures to enhance the overall presentation.
Answer: Thank you for pointing this out- we have changed the font of the appendix to match the rest of the text.
Reviewer 3 Report
Comments and Suggestions for Authors
I find the objectives of the study highly interesting and relevant to the literature. However, there are several aspects of the study that need improvement to validate the proposed methodology.
Line 43:
Please provide a detailed description of the methodologies used to analyze fruit and vegetable consumption and how these are included in the assessment of community-based programs. It is essential to mention the methodologies (e.g., 24-Hour Recall, Frequency Questionnaire) and their barriers and challenges in implementation. This will help justify how your methodology advances the current state of the art.
Line 49:
I would suggest mentioning that these studies are particularly relevant for families with children, as the negative impact of low fruit and vegetable consumption and high intake of unhealthy products has been demonstrated. For example:
Reales-Moreno M, Tonini P, Escorihuela RM, Solanas M, Fernández-Barrés S, Romaguera D, Contreras-Rodríguez O. Ultra-Processed Foods and Drinks Consumption Is Associated with Psychosocial Functioning in Adolescents. Nutrients. 2022 Nov 15;14(22):4831. doi: 10.3390/nu14224831. PMID: 36432518; PMCID: PMC9694351.
Line 90:
Since you mention that the methodology was developed through a co-creation process, you should provide more details about how many people were involved and the nature of their involvement (e.g., did they contribute to the entire questionnaire or only specific sections?).
Line 123:
Regarding fruit and vegetable consumption, I appreciate that you included category descriptions to clarify for respondents what is considered as fruit and vegetables. However, there are several limitations that should be addressed:
- For consumption, I’m not sure if "times" is the correct unit of measurement; instead, I suggest using "portions."
- To clarify what a portion is, you could include an image as a reference unit (e.g., one plate of vegetables equals one portion; a side salad equals half a portion). A portion is approximately 80g, considering the UN’s recommendation of 400g per day (four portions).
- Additionally, I recommend including only fresh vegetables in the analysis and excluding frozen ones.
Line 152:
Please explain how you analyzed the qualitative section of the questionnaire. This is crucial to ensure a standardized methodology that can be replicated in different contexts. I suggest using software and specifying how the data was extracted.
Line 235:
Since your hypothesis was not supported due to the small sample size, I recommend applying a methodology to increase the data volume based on the patterns identified in your sample. Specifically, I suggest using bootstrapping and adding statistical perturbation to make your data more robust. Another option could be leveraging data from other projects and testing them within your framework.
Author Response
Line 43:
Please provide a detailed description of the methodologies used to analyze fruit and vegetable consumption and how these are included in the assessment of community-based programs. It is essential to mention the methodologies (e.g., 24-Hour Recall, Frequency Questionnaire) and their barriers and challenges in implementation. This will help justify how your methodology advances the current state of the art.
Answer: We have added a more detailed explanation and citations that explicitly describe other fruit and vegetable consumption tools.
“Tools used to assess fruit and vegetable intake generally include a food frequency questionnaire (FFQ) consisting of asking the consumption frequency of up to 120 food items, a 24-hour recall, or diet records [10, 11]. These tools are not widely applied in di-verse community settings as they are difficult to implement and have barriers to participation. Outside of rigorous studies with few participants, high costs, and low accessibility, there is a need for more innovative approaches to measure the effectiveness of programs [12].”
Line 49:
I would suggest mentioning that these studies are particularly relevant for families with children, as the negative impact of low fruit and vegetable consumption and high intake of unhealthy products has been demonstrated. For example:
Reales-Moreno M, Tonini P, Escorihuela RM, Solanas M, Fernández-Barrés S, Romaguera D, Contreras-Rodríguez O. Ultra-Processed Foods and Drinks Consumption Is Associated with Psychosocial Functioning in Adolescents. Nutrients. 2022 Nov 15;14(22):4831. doi: 10.3390/nu14224831. PMID: 36432518; PMCID: PMC9694351.
Answer: Thank you for mentioning this important reference. We have mentioned it in the literature review.
“Previous research and firsthand accounts have revealed common lived experiences including the need for improved accessibility, affordability, quality, and healthier options, especially for families with children or adolescents [13–15].”
Line 90:
Since you mention that the methodology was developed through a co-creation process, you should provide more details about how many people were involved and the nature of their involvement (e.g., did they contribute to the entire questionnaire or only specific sections?).
Answer: We added a sentence explaining that program staff and experts were able to review and give input on the entire questionnaire as well as the collection methods.
“Program staff and experts were able to suggest and review the entire questionnaire and collection methods during the development stage and community members were asked about their thoughts at the time of collection.”
Line 123:
Regarding fruit and vegetable consumption, I appreciate that you included category descriptions to clarify for respondents what is considered as fruit and vegetables. However, there are several limitations that should be addressed:
- For consumption, I’m not sure if "times" is the correct unit of measurement; instead, I suggest using "portions."
- To clarify what a portion is, you could include an image as a reference unit (e.g., one plate of vegetables equals one portion; a side salad equals half a portion). A portion is approximately 80g, considering the UN’s recommendation of 400g per day (four portions).
- Additionally, I recommend including only fresh vegetables in the analysis and excluding frozen ones.
Answer: We agree that it is challenging to select the correct unit of measure in a survey question. We gave this particular question a lot of thought and review and ultimately decided to align with the Behavioral Risk Factor Surveillance System for a comparable data source. There are pros and cons to the different ways of asking, and this could be an area of future research. The main goal of this tool was to make it as brief yet comprehensive as possible, so we included a similar description of what constitutes a fruit and vegetable to the BRFSS. This assumes that fresh and frozen vegetables have similar nutritional characteristics. During this pilot study, we were not able to collect data on all areas of interest, including this difference.
Line 152:
Please explain how you analyzed the qualitative section of the questionnaire. This is crucial to ensure a standardized methodology that can be replicated in different contexts. I suggest using software and specifying how the data was extracted.
Answer: We have added a paragraph to the data analysis section in the methods explaining how we looked at qualitative data specifically.
“Qualitative data, primarily from the open text responses, was categorized manually due to the small data size and focus on quantitative data. Comments were categorized by positive and negative sentiment, comments regarding the program delivery, and comments regarding the survey or collection method itself.”
Line 235:
Since your hypothesis was not supported due to the small sample size, I recommend applying a methodology to increase the data volume based on the patterns identified in your sample. Specifically, I suggest using bootstrapping and adding statistical perturbation to make your data more robust. Another option could be leveraging data from other projects and testing them within your framework.
Answer: Thank you for this suggestion. Due to the small sample size and that this is a pilot study, we decided not to apply any statistical techniques such as bootstrapping to the data. We did want to compare the findings to other comparable data sources, such as BRFSS, but this was only a sensitivity analysis to explore if the association was worth implementing a full data collection effort.
Round 2
Reviewer 2 Report
Comments and Suggestions for Authors
Dear Authors,
I appreciate your efforts in improving the article and including my content-related remarks.
So, I accept this paper in the present form.
Regards,
Reviewer 3 Report
Comments and Suggestions for Authors
No more comments from my side